# Diversity and Identification of Plant-Parasitic Nematodes in Wheat-Growing Ecosystems

**DOI:** 10.3390/microorganisms10081534

**Published:** 2022-07-29

**Authors:** Ayça İrem Keçici, Refik Bozbuğa, Atilla Öcal, Ebubekir Yüksel, Göksel Özer, Şenol Yildiz, Rachid Lahlali, Brigitte Slaats, Abdelfattah A. Dababat, Mustafa İmren

**Affiliations:** 1Department of Plant Protection, Faculty of Agriculture, Bolu Abant Izzet Baysal University, Bolu 14030, Türkiye; ayca.kecici@icloud.com (A.İ.K.); gokozer@gmail.com (G.Ö.); s7yildiz@gmail.com (Ş.Y.); m.imren37@gmail.com (M.İ.); 2Department of Plant Protection, Faculty of Agriculture, Eskisehir Osmangazi University, Eskisehir 26160, Türkiye; refikbozbuga@gmail.com; 3Atatürk Horticultural Central Research Institute, Yalova 77100, Türkiye; atilla_ocal@hotmail.com; 4Department of Plant Protection, Faculty of Agriculture, Kayseri Erciyes University, Kayseri 38030, Türkiye; ebubekiryuksel@erciyes.edu.tr; 5Phytopathology Unit, Department of Plant Protection, Ecole Nationale d’Agriculture de Meknès, Km10, Rte Haj Kaddour, B.P. S/40, Meknes 50001, Morocco; rlahlali@enameknes.ac.ma; 6EAME Technical Expert Seedcare, Syngenta Crop Protection AG Office, Rosentalstr 67, 4058 Basel, Switzerland; brigitte.slaats@syngenta.com; 7International Maize and Wheat Improvement Centre (CIMMYT), Ankara 06170, Türkiye

**Keywords:** ITS region, plant-parasitic nematodes, Sakarya, wheat

## Abstract

Several nematode species can be found in different densities in almost any soil ecosystem, and their diversity in those ecosystems depends on numerous reasons, such as climatic conditions and host presence. Cereals are one of the main hosts of plant-parasitic nematodes (PPN), chiefly root-lesion nematodes (RLN, *Pratylenchus* spp.) and cereal cyst nematodes (CCN, *Heterodera* spp.). These nematodes are known as major parasites of the cereal crops; however, agricultural areas accommodate various nematodes showing biological variation. The diversity of parasitic nematodes on cereals in the Sakarya provinces of Türkiye, where cereals are intensively grown and located in the middle of two climatic zones, has not been well studied. Therefore, in this study, we aimed to determine the diversity, identification, and molecular phylogeny of PPNs in wheat-growing ecosystems in the Hendek, Pamukova, Geyve, Akyazı, and Central districts of Sakarya. The diversity of PPNs was calculated using the Shannon diversity index. Thirteen PPN genera were detected in 92% of soil samples. *Heterodera filipjevi* was identified in 24% of the soil samples using morphological, morphometrical, and molecular tools. In the morphological and molecular analyses, intraspecific polymorphism was observed in *H. filipjevi* populations. The result indicated that the high infestation rate of *H. filipjevi* was recorded from Geyve and Pamukova, followed by Hendek and Akyazı; however, a low infestation rate was detected in the Central district. The moderate value of the Shannon index of migratory nematode species was obtained in wheat fields as 2.31, whereas the value of evenness was 0.93, implying moderate diversity and high evenness of nematodes. This study is the first comprehensive report on *H. filipjevi* from wheat cropping areas in the Sakarya province. Intensified cereal cropping systems with/without non-cereal rotations increased the risk of plant-parasitic nematodes, especially RLNs and *H. filipjevi* infection of wheat production areas in the province.

## 1. Introduction

Türkiye is one of the world’s top 10 wheat producers, with 20 million tonnes of average annual wheat production [1]. Wheat production in Türkiye climbed from 10.7 thousand tons in 1971 to 18.2 thousand tons in 2020, expanding at an annual increase of 1.85% yearly pace on average [2]. However, wheat yield is highly variable among the different geographical zones of Türkiye, primarily because of climatic conditions and poor management methods of plant disease and pests [3]. In 2021–2022, the wheat planting area dropped to 7.05 million hectares, with a total production of 17.6 million tons, which is insufficient to meet Türkiye’s wheat industry’s needs [2].

Fungus and nematode-induced soil-borne diseases are among the main constraints in agricultural production and are quite challenging to be managed. Plant-parasitic nematodes alone are responsible for nearly 10% of yield losses worldwide, resulting in economic losses of over USD 125 billion per year [4,5]. However, yield losses associated with PPNs generally go unnoticed by a great proportion of farmers due to the soil-dwelling nature of most PPNs and the similarity of symptoms with other biotic and abiotic disease-inducing factors. Even though it is established that the problem is nematode-originated, it is still quite difficult to differentiate PPN species due to the co-occurrence of different species of PPNs in the field and the complexity of diagnostic characteristics. Among the PPNs that cause a reduction in the yield and profitability of wheat production, Cyst (*Heterodera* spp.), Root-lesion (*Pratylenchus* spp.), Root-knot (*Meloidogyne* spp.), Seed-gall (*Anguina* spp.), and Stubby-root nematodes (*Paratrichodorus* spp.) are the most economically important genera [6].

The sedentary endoparasites are the most injurious and economically destructive groups attacking cereals [6,7]. Cyst nematodes have been extensively investigated as a category of endoparasitic nematodes as they lead to substantial yield losses. Numerous research studies have been conducted to clarify the biology of cyst nematodes concerning their host plants [8,9]. Cyst nematodes, such as *Heterodera*, infect various host plants, including wheat [10,11]. This genus has 12 known species with overlapping morphological features, making it difficult to separate them, especially when they occur in the same niche [12,13]. Molecular diagnostic tools enable researchers to correctly and rapidly identify morphologically indistinct PPN species. The internal transcribed spacer sequences (ITS) region, including non-coding ITS1 and ITS2, and the 5.8S region of ribosomal DNA (rDNA), have proved very beneficial in accurately identifying and characterizing PPN species [14,15].

In Türkiye, *Heterodera* spp. populations obtained from different wheat cropping systems have mainly been identified based on their morphology/morphometrics and molecular tools [16,17,18,19,20]. Although Sakarya is one of the significant wheat-producing provinces of Türkiye in the Marmara region, our knowledge is still insufficient regarding the morphometrics and genetics of CCNs, including the diversity and variability of CCN. Thus, the specific objectives of this study were to (i) investigate the occurrence and prevalence of the important plant-parasitic nematode genera in the Marmara region, (ii) identify both mature cysts and second-stage juveniles (J2s) of *H. filipjevi* group using morphological/morphometrical and molecular tools, and (iii) examine polygenetic relationships of the collected populations.

## 2. Materials and Methods

Soil samples were collected from Hendek, Pamukova, Geyve, Akyazı, and the Central districts of monoculture cereal production areas in Sakarya province in 2018 (Figure 1). Surveys were performed during wheat’s heading and flowering times for migratory nematodes and at harvesting time for the sedentary nematodes. A total of 40 soil samples were randomly collected from each location for each sampling time (Table 1).

A modified Baermann funnel technique was employed to recover motile nematodes from 100 cm^3^ of soil per soil sample [21]. The extracted nematodes were transferred to cylinder measures and allowed to sink and settle down at the bottom for 8 h before pouring the settled nematodes into 15 mL tubes. Migratory nematodes were counted and identified to the genus level using a light microscope (DM1000, Leica Microsystems, Wetzlar, Germany) at 100× magnification [22]. 

The modified sieving–decanting method was used to remove sedentary nematodes (*Heterodera* spp.) from 200 cm^3^ of soil using 60 mesh (0.25 mm) and 100 mesh (0.015 mm) sieves, as described by [21,22]. From each sample, at least ten healthy cysts were hand-picked and kept at 4 °C for genetic and morphological examinations. Sedentary nematodes were morphologically identified to the genus level under a stereomicroscope (Zeiss Stemi 305, Carl Zeiss, Jena, Germany). The incidence of the nematode (number of samples with nematode/total number of samples) was computed for each sample location [23]. 

Second-stage juveniles and males were relaxed in a water bath at 65 °C for 5 min, fixed in a mixture of ethanol, acetic acid, and formalin (20:6:1), processed into glycerine solution, and mounted on permanent slides with anhydrous glycerin and paraffin seal [23,24]. Measurements were taken for the following morphological structures: length, width at the midbody, stylet length, labial region height and width, dorsal oesophageal gland to spear knobs base, anterior end to the median bulb, anterior end to excretory pore, oesophageal length, width at the anus and hyaline region, tail length, hyaline tail length, and spicules length (males). To see the vulval cone, cysts were soaked in 45% of lactic acid for 15 min, transferred to water, dissected, and mounted in glycerin with a paraffin seal for viewing and analysis [23,24,25]. The vulval slit length, fenestral length, and breadth were all measured. All measurements were measured under a light microscope (CX31, Olympus, Tokyo, Japan) equipped with the Infinity Analyze software version 6.5.2 (Lumenera Inc., Ottawa, ON, Canada) [26,27]. 

The data were analyzed using analysis of variance (ANOVA) techniques in the SPSS 10.0 software for Windows (SPSS Inc., Chicago, IL, USA) to evaluate if there were any statistically significant differences between the populations (*p* ≤ 0.05). The standard test of means was used to detect if the variance across populations is statistically significant.

DNA extraction and PCR reaction to amplify the ITS of ribosomal DNA using the Direct PCR Master kit (Cat. No. PCR-111S; Jena Bioscience GmbH, Jena, Germany) following the manufacturer’s recommendations. A single cyst from each sample was used for direct PCR amplification via the Direct PCR Master kit. PCR reaction mix consisted of 1 of sample lysate, 2 µL of each 10 µM forward primer AB28 (5ʹ-CGTAACAAGGTAGCTGTAG-3ʹ), and reverse TW81 (5ʹ-TCCTCCGCTAAATGATATG3ʹ) (28); 25 µL Direct PCR Master Mix, and PCR-grade water up to 50 µL [25]. PCR conditions included initial denaturation at 94 °C for 4 min, 35 cycles at 94 °C for 1 min, 55 °C for 1.5 min, 72 °C for 2 min, and final elongation at 72 °C for 10 min in a T100 thermal cycler (Bio-Rad, Hercules, CA, USA) [28,29]. 

The amplicons were purified and sequenced in both directions using the same primers by a commercial sequencing company (Macrogen Inc., Seoul, Korea). The DNA sequences were edited, and consensus sequences were computed manually using Mega X computing platforms [30]. All sequences were compared against the GenBank database, National Center for Biotechnological Information (NCBI), using the BLASTn algorithm and deposited in GenBank. The obtained isolates from this study plus additional sequences retrieved from the GenBank database were aligned in the MAFFT v.7 online interface using default settings and manually edited with MEGA X. A maximum likelihood (ML) tree of ITS data set was inferred using the command-line version of IQ-TREE 1.6.7 with ultrafast bootstrapping implemented with 1000 replicates. Analyses were run on the CIPRES Science Gateway V 3.3. [30,31,32,33]. The resulting trees were analyzed and edited in FigTree v1.4.2 software. The ITS sequence of *Globodera pallida* (HQ670281) was included as an outgroup to facilitate the production of consensus trees.

The Shannon Diversity Index is a method to quantify species diversity in a community. It is calculated with the formula: H = −Σpi × ln(pi) that Σ: A Greek symbol that means “sum”; ln: Natural log; pi: The proportion of the entire community made up of species “I”. A higher value of H represents a higher diversity of species in a specific community, while a lower value of H indicates a lower diversity within a particular community. The Shannon Equitability Index is a method to measure the evenness of species in a community [30]. The term “evenness” (EH) signifies how related the abundances of different species are in the community [30].

EH is calculated as: EH = H/ln(S), where H is the Shannon Diversity Index and S is the total number of unique species. This value ranges from 0 to 1, where 1 signifies whole evenness [33,34]. Shannon diversity index (H) is classified based on the subsequent grouping: low (H < 2), moderate (2 < H < 4), and high (H > 4) species of gastropods [34].

## 3. Results

### 3.1. Occurrence of Plant-Parasitic Nematodes

The genera of PPNs; *Helicotylenchus* Steiner 1945, *Heterodera* Schmidt 1871, *Merlinius* Siddiqi 1970, *Pratylenchoides* Winslow 1958, *Pratylenchus* Filipjev 1936, *Trophurus* Loof 1956, *Paratrophurus* Arias 1970, *Filenchus* Andrassy 1954, *Tylenchus* Bastian 1865, *Scutylenchus* Jairajpuri 197, *Amplimerlinius* Siddiqi 1976, *Boleodorus* Thorne 1941, and *Basiria* Siddiqi 1959 were detected in the surveyed areas (Table 2). The PPN species were found in 92% of soil samples collected from Hendek, Pamukova, Geyve, Akyazı, and the Central districts of Sakarya province. The highest occurrence was detected in the genus of *Tylenchus*, having a frequency of 42% in the Geyve district, followed by *Filenchus,* having a frequency of 25% in the Taraklı district (Table 3). The lowest frequency was found for the genus of *Boleodorus* and *Basiria* with 5%. The economically important PPN genera detected in the surveyed areas were *Pratylenchus*, *Heterodera*, *Helicotylenchus*, *Merlinius*, *Trophurus*, *Paratrophurus*, *Pratylenchoides*, and *Amplimerlinius* (Table 3) [35,36]. 

The RLNs, *Pratylenchus* spp. populations were found in 45% of the soil samples collected from wheat fields in 18 different locations of Sakarya Province (Table 3). The nematodes were recorded from three, four, two, and five fields located in Hendek, Pamukova, Geyve, and Akyazı, respectively. Geyve has the lowest nematode occurrence rate in infested fields. The highest density of nematodes was 25 juveniles or mature nematodes per g of soil, which was obtained from Pamukova, while the lowest nematode density was detected in samples taken from Hendek (Table 4).

The cyst nematodes were found in 30% of wheat fields in Geyve, Taraklı, Söğütlü and Kaynarca in Sakarya Province. All *Heterodera* populations were identified as *H. filipjevi* based on both morphological and molecular analysis. *Heterodera filipjevi* was found mostly in fields where monoculture systems are practiced, such as Geyve, Taraklı, and Kaynarca districts (Table 5). High cyst incidence levels were recorded in the fields of Kaynarca and Taraklı districts. The lowest infested fields were reported from the Söğütlü district, where only one out of four fields was infested. The highest number of cysts was found in the Taraklı district with nine cysts, whereas the lowest number of two cysts was found in samples from the Central district (Table 5). 

### 3.2. Morphological Measurements

Twelve *H. filipjevi* populations were identified in the samples from Geyve, Taraklı, and Central districts (Table 5). Morphometric and morphological characteristics have close similarities with those described by [37]. Lemon-shaped cysts with a posterior protuberance, the vulval cone was bifurcate with a horseshoe-shaped semifenestra and prominent bullae and underbridge (Figure 2). Cysts (*n* = 10) were measured for the following dimensions (range, mean, SD): body length without neck 780 µm (from 670 to 862 µm), neck length 95 µm (from 76 to 116 µm), fenestra length 52 µm (from 42 to 60 µm), and breadth 24.5 µm (from 21 to 27 µm) (Table 6). Juveniles in their second stage had a cylindrical form with a slightly offset head and a tapering circular tail tip. The stylet was robust, with shallow concave basal knobs on the front side (Figure 2). *Heterodera filipjevi* juveniles had a body length of 477–516 µm and a stylet length of 22–25 µm (Table 6), with fairly concave stylet knobs. Lateral fields were divided into four different lines; however, only the two inner lines were sometimes distinct. 

Shannon diversity index (H) is classified based on the subsequent grouping: low (H < 2); moderate (2 < H < 4); and high (H > 4) species of gastropods [34]. In this study, the Shannon diversity index (H) was calculated as H = 2.31767 in 12 migratory nematode species in wheat fields. The value of the H index is between 2 and 4, which means moderate diversity of nematode genus. The evenness value was calculated as EH = 0.932699. These results showed that nematode evenness was high in wheat-planted areas because EH is close to 1 (Table 7).

### 3.3. Molecular Features

The sequences of the ITS rRNA gene recovered from the Sakarya Province population of *H. filipjevi* (Sak_Hf11) varied by 1 bp from other *H. filipjevi* population. Figure 3 depicts the phylogenetic connections of Sakarya populations of *H. filipjevi* to other populations derived from GenBank and species in the Avenae group. The phylogenetic relationships of *H. filipjevi* populations were compared to sequences from other countries, including China, the USA, Italy, and Türkiye. The phylogenetic tree was constructed using 1000 bootstrapped sequence alignments randomly replicated globally. The *H. filipjevi* sequences constitute a single clade within the tree. 

Results indicated a clear separation of the cyst nematode, *H. filipjevi* confirmed the link between genotyping and phenotyping traits. Any intraspecific polymorphism was not found among *H. filipjevi* populations placed in the same phylogenetic group and contributed to GenBank, as indicated by high bootstrap values (Figure 3). All accession numbers provided by GenBank for the populations obtained in this study are shown in Figure 3.

## 4. Discussion

The results indicated that plant-parasitic nematodes were found in 92% of soil samples, and 13 taxa of nematodes were recovered from cereal roots and soil samples. *Helicotylenchus* spp., *Heterodera* spp., *Merlinius* spp., *Pratylenchoides* spp., *Pratylenchus* spp., *Trophurus* spp., *Paratrophurus* spp., *Filenchus* spp., *Tylenchus* spp., *Scutylenchus* spp., *Amplimerlinius* spp., *Boleodorus* spp., and *Basiria* spp. were detected in the surveyed areas. Apart from *Pratylenchoides sheri*, *Merlinus brevidens*, *Amplimerlinus vicia*, *Paratrophurus striatus*, and *P. acristylus*, the most prevalent genera of PPNs were discovered previously in wheat-producing regions of Türkiye’s Southeast Anatolian region [38]. *Merlinius brevidens*, *Scutylenchus quadrifer*, *H. latipons*, *P. thornei*, and *Aphelenchus avenae* were the most frequently discovered PPNs in the cereal cropping system of Adıyaman region in Türkiye [39]. Consequently, the identified genera are economically significant for grain production in Sakarya Province. 

The findings of this study revealed that there is an intensive presence of cereal cyst nematode species *H. filipjevi* in cereal growing locations (wheat and barley areas as main crops) from Geyve, Taraklı, Söğütlü, and Kaynarca districts. The results suggested that 24% of the soil samples were infested with *H. filipjevi* in wheat production areas in Sakarya Province. The fields where cyst nematodes were not detected were generally rotated with other non-cereal crops. However, *H. filipjevi* was found mostly in monoculture system in Geyve, Taraklı, Söğütlü, and Kaynarca districts (Table 3). High cyst incidence levels were recorded, e.g., four out of five and three out of four fields were infested in Kaynarca and Taraklı districts, respectively. Söğütlü district has the lowest prevalence of infested fields, with just one out of every four fields affected. In an earlier study, the most extensively dispersed species were *H. filipjevi*, *H. latipons*, *P. thornei*, and *P. neglectus* in the Anatolia region of Türkiye [18]. Another study demonstrated that 56% of wheat fields in Elazig, Malatya, Sivas, Erzurum, Erzincan, Igdir, and Kars provinces in Türkiye were infected with *H. filipjevi* [36]. Imren et al. [40] observed that 83% of wheat fields in Bolu province, Türkiye, were infected with the cereal cyst nematode, *H. filipjevi*. 

Toktay et al. reported that the most prevalent phytophagous nematodes in cereal soils in Niğde province were those belonging to the genera *Heterodera*, *Ditylenchus*, *Merlinius*, *Pratylenchus*, *Aphelenchus*, *Aphelenchoides*, *Tylenchus*, *Helicotylenchus*, *Trophurus*, *Pratylenchoides*, *Filenchus*, and *Xiphinema*. [41]. It suggests that the presence of climatic circumstances conducive to *H. filipjevi* completing its life cycle may be a significant element in presenting a danger to grain output in Sakarya Province.

The findings of this current study suggested that there is no discernible variability in *H. filipjevi* populations based on morphological variables, confirming that cyst and J2 body measurements vary. *H. filipjevi* is a close relative of *H. avenae*, with minor morphological differences separating them [36,37]. *H. filipjevi* has smaller bullae and a noticeable, though narrow, underbridge. *H. latipons*, on the other hand, was identified from *H. filipjevi* by its strong underbridge and lack of conspicuous bullae in the vulval cone. The absence of an underbridge and the presence of bullae around the vulval cone of *H. filipjevi* cysts are acknowledged as useful physical characteristics for identification. The vulval cones of *H. filipjevi* and *H. latipons* are readily differentiated by an underbridge and bullae in *H. latipons* [13,42,43]. In comparison to *H. latipons*, *H. filipjevi* second-stage juveniles have a longer tail, stylet, and hyaline section of the tail [42,43]. 

The Shannon diversity index estimates the diversity of species within a community and shows how diverse the species in a particular community are and their relative abundance [34]. The Shannon biodiversity index of migratory nematodes in wheat fields was moderate. This implies that wheat affects on nematode community. It is expected that nematode richness in the soil would be high in usually undisturbed ecosystems by human beings. This situation suggests that it may be related to monoculture agriculture (Table 7).

DNA data to determine phylogeny are well established for various groups of animals. Numerous articles have shown that molecular data may be a very significant resource for cyst nematode systematics [44,45,46,47,48]. The genetic variety of species, both inter- and intra-specific, has been found to be beneficial in identifying the nematode phylogeny. The use of molecular approaches in combination with the analysis of ribosomal RNA gene (rRNA) sequences has bolstered our confidence in our understanding of cyst nematode relationships [49,50]. Ribosomal RNA genes are one of the most well-characterized gene families in worms [51,52]. While the majority of the RNA genes are highly conserved, there is significant diversity in discrete parts of the genes, as well as the length and sequence of the spacer regions. Unlike their spacer sections, the ribosomal genes change slowly but retain some useful genetic information.

Typically, the rRNA array consists of three ribosomal genes, 18S, 5.8S, and 28S, which are linked in repeating units together with their spacer segments, ITS1 and ITS2. This work demonstrated the significance of certain cyst and J2 body dimension traits in identifying various populations of *H. filipjevi* by comparing them to rDNA sequences. Molecular and morphological evidence indicate that species within the *H. filipjevi* population may be classified as a single group. Similarly, Bekal et al. [38] and Imren et al. [40] found little genetic diversity amongst *H. filipjevi* populations of Türkiye’s Mediterranean area. Subbotin et al. [42], on the other hand, discovered intraspecific polymorphism within the *H. filipjevi* population. Toktay et al. [1] have also reported the presence of intraspecific diversity in Türkiye’s Eastern Anatolian area of *H. filipjevi* population.

## 5. Conclusions

The main goal of this research was to identify the eco-regional distribution of the major genera of plant-parasitic nematodes in Sakarya Province, with a focus on the cereal cyst nematode species *H. filipjevi*. In conclusion, this survey makes the following recommendations to local technical personnel and researchers: diversify wheat cultivars to include durum wheat in areas with high cyst counts, as resistant durum wheat than spring wheat; apply cultural practices, especially crop rotation practices; breeding for germplasms with high resistance potentials against cereal cyst nematodes; and finally, educate more technical staff to work on soil-borne disease topics in the region. Additional extensive surveys in Sakarya province and comprehensive pathotype investigations of *H. filipjevi* are still required to draw a suitable control strategy.

## Figures and Tables

**Figure 1 microorganisms-10-01534-f001:**
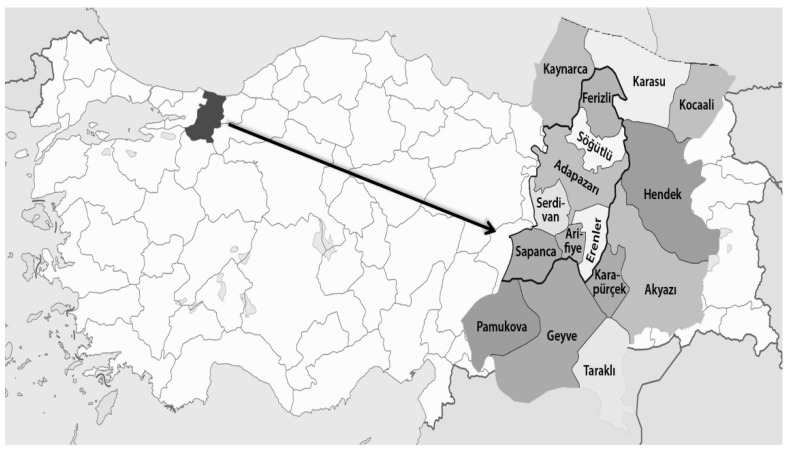
Surveyed locations in the Sakarya province for plant-parasitic nematodes.

**Figure 2 microorganisms-10-01534-f002:**
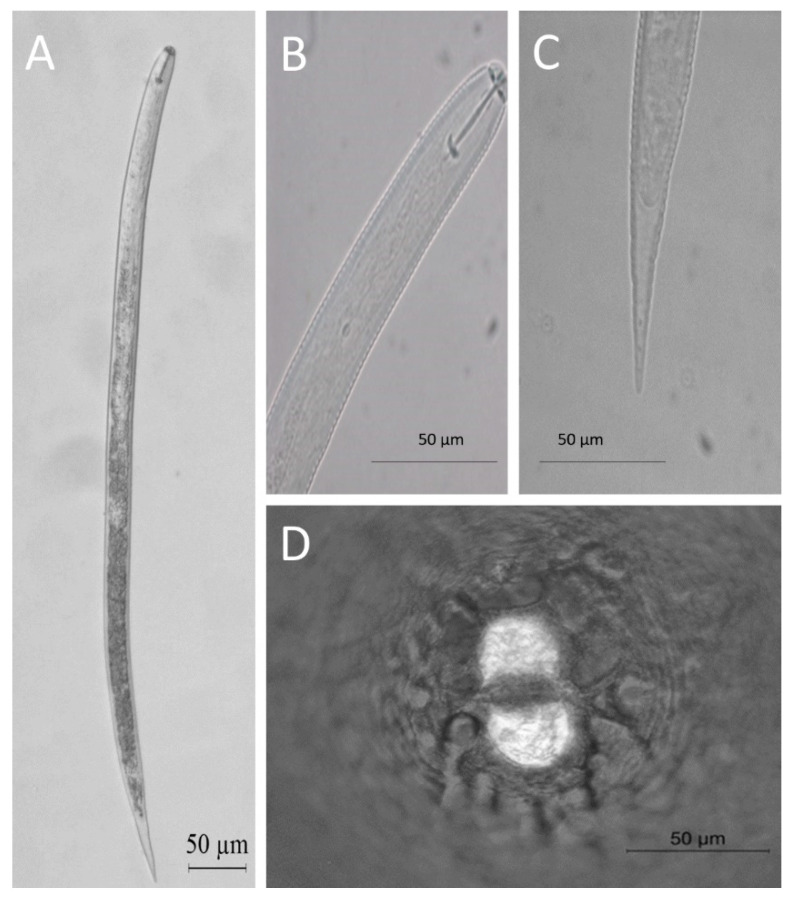
*Heterodera filipjevi* image under light microscope: (**A**) second stage juveniles; (**B**) head region; (**C**) tail region; (**D**) fenestral region.

**Figure 3 microorganisms-10-01534-f003:**
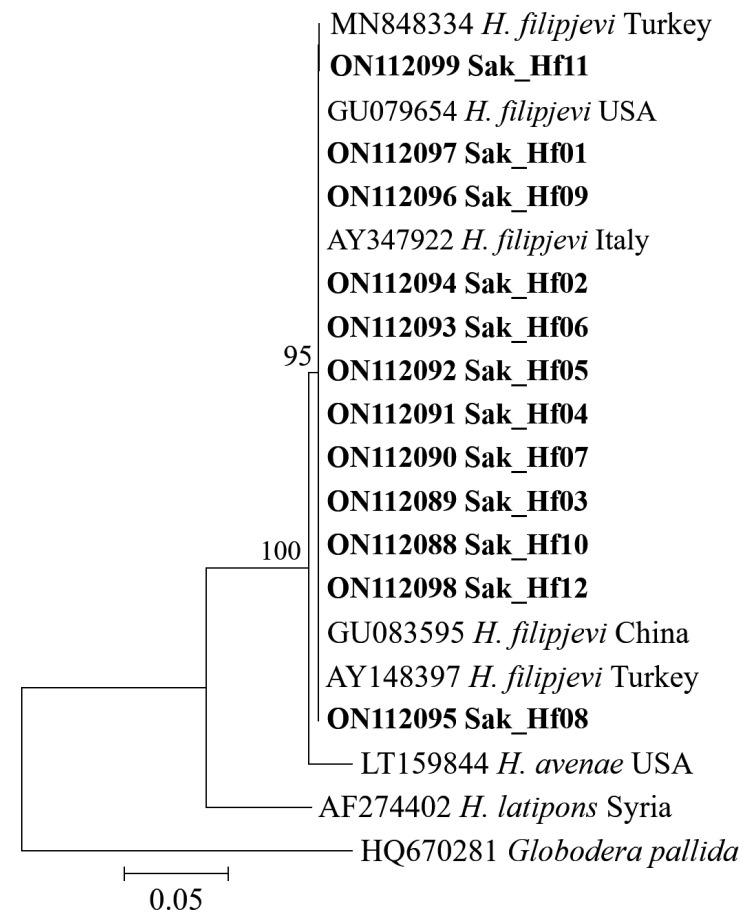
Phylogenetic tree based on maximum-likelihood using IQ-TREE ITS sequences of *Heterodera* populations. The bootstrap values (left) are at each node. Isolates obtained in this study are indicated in bold.

**Table 1 microorganisms-10-01534-t001:** The locations of the plant-parasitic nematodes collected from different wheat fields in the Sakarya province.

No.	District	Location	Latitude	Longitude
1	Hendek	Beylice	40°83′13″	30°84′80″
2		Beylice	40°83′04″	30°84′24″
3		Bıçkıatik	40°83′85″	30°85′69″
4		Beylice	40°83′85″	30°88′34″
5		Beylice	40°83′28″	30°86′14″
6	Ferizli	Karadiken	40°96′13″	30°47′70″
7		Sebiller	40°96′13″	30°43′25″
8		Doğancı	40°98′44″	30°43′06″
9		Karadiken	40°95′59″	30°51′48″
10		Karadiken	40°96′15″	30°43′27″
11	Kaynarca	Kızılcaali	40°98′66″	30°40′71″
12		Kızılcaali	40°98′27″	30°40′58″
13		Merkez	41°03′33″	30°30′77″
14		Küçükkaynarca	41°03′38″	30°32′48″
15	Geyve	Merkez	40°51′81″	30°30′91″
16		Merkez	40°51′78″	30°30′94″
17		Merkez	40°51′74″	30°31′06″
18		Merkez	40°50′41″	30°30′55″
19		Merkez	40°51′18″	30°29′81″
20		Ilıca	40°45′25″	30°38′10″
21		Çamlık	40°45′24″	30°37′93″
22	Taraklı	Hacıyakup	40°44′69″	30°37′94″
23		Hacıyakup	40°43′95″	30°43′85″
24		Hacıyakup	40°43′98″	30°43′90″
25		Aksu	40°44′64″	30°45′37″
26		Aksu	40°44′62″	30°46′51″
27		Aksu	40°44′52″	30°46′59″
28		Aksu	40°44′37″	30°47′00″
29	Söğütlü	Merkez	40°44′28″	30°43′80″
30		Merkez	40°44′13″	30°43′91″
31		Merkez	40°86′42″	30°49′01″
32	Akyazı	Merkez	40°89′69″	30°46′43″
33		Merkez	40°67′62″	30°62′37″
34		Şerefiye	40°61′73″	30°67′32″
35		Şerefiye	40°61′68″	30°67′43″
36		Merkez	40°05′33″	30°85′00″
37	Pamukova	Merkez	40°05′32″	30°85′06″
38		Mekece	40°45′44″	30°04′85″
39		Mekece	40°45′51″	30°04′72″
40		Mekece	40°45′48″	30°04′28″

**Table 2 microorganisms-10-01534-t002:** The plant-parasitic nematodes surveyed wheat areas in Sakarya province.

No.	Districts	Locations	Nematode Genera
1	Hendek	Beylice	*Pratylenchus*; *Dorylaimida*; *Basiria*
2		Beylice	*Ditylenchus*
3		Bıçkıatik	*Dorylaimida*
4		Beylice	*Dorylaimida*
5		Beylice	*Dorylaimida*
6	Ferizli	Karadiken	*Scutylenchus*; *Filenchus*; *Dorylaimida*
7		Sebiller	*Helicotylenchus*; *Dorylaimida*
8		Doğancı	*Helicotylenchus*; *Scutylenchus*; *Filenchus*
9		Karadiken	*Dorylaimida*
10		Karadiken	*Filenchus*
11	Kaynarca	Kızılcaali	*Dorylaimida*
12		Kızılcaali	*Filenchus*; *Aphelenchus*
13		Merkez	*Helicotylenchus*; *Scutylenchus*; *Dorylaimida*
14		Küçük Kaynarca	*Tylenchus*; *Dorylaimida*
15	Geyve	Town center	*Merlinius*; *Filenchus*; *Ditylenchus*
16		Town center	*Pratylenchus*; *Scutylenchus*
17		Town center	*Pratylenchus*
18		Town center	*Scutylenchus*; *Filenchus*; *Dorylaimida*
19		Town center	*Ditylenchus*; *Dorylaimida*
20		Ilıca	*Pratylenchus*; *Paratrophurus*; *Filenchus*; *Ditylenchus*; *Dorylaimida*
21		Çamlık	*Pratylenchus*; *Merlinius*; *Dorylaimida*
22	Taraklı	Hacıyakup	*Merlinius*; *Filenchus*; *Dorylaimida*
23		Hacıyakup	*Merlinius*; *Pratylenchus*; *Dorylaimida*
24		Hacıyakup	*Ditylenchus*
25		Aksu	*Scutylenchus*; *Filenchus*
26		Aksu	*Filenchus*
27		Aksu	*Ditylenchus*; *Tylenchus*; *Boleodorus*
28		Aksu	*Helicotylenchus*; *Filenchus*; *Ditylenchus*
29		Hacıyakup	*Pratylenchus*; *Merlinius*; *Ditylenchus*; *Scutylenchus Filenchus*
30		Hacıyakup	*Dorylaimida*
31	Söğütlü	Mahsudiye	*Pratylenchus*; *Aphelenchoides*; *Filenchus*; *Boleodorus*; *Dorylaimida*
32		Town center	*Ditylenchus*; *Dorylaimida*
33	Akyazı	Town center	*Helicotylenchus*; *Amplimerlinius*; *Merlinius*; *Ditylenchus*; *Filenchus*; *Dorylaimida*
34		Şerefiye	*Helicotylenchus*; *Ditylenchus*; *Monachus*; *Dorylaimida*
35		Şerefiye	*Aphelenchoides*; *Ditylenchus*; *Dorylaimida*
36	Kocaali	Town center	*Ditylenchus*; *Amplimerlinius*; *Scutylenchus*; *Monachus*; *Scutylenchus*
37		Town center	*Helicotylenchus*; *Ditylenchus*; *Scutylenchus*; *Filenchus Dorylaimida*
38	Pamukova	Mekece	*Merlinius*; *Ditylenchus*; *Filenchus*; *Dorylaimida*
39		Mekece	*Helicotylenchus*; *Dorylaimida*
40		Mekece	*Merlinius*; *Ditylenchus*; *Tylenchus*

**Table 3 microorganisms-10-01534-t003:** Migratory nematode genera recovered from the sampled fields in Sakarya province.

No.	Nematode Genera	Infested Fields (%) *	Nematode Abundance **
1	*Merlinus*	28	960 ± 120 (220–1100)
2	*Trophurus*	16	240 ± 20 (200–360)
3	*Paratrophurus*	14	140 ± 40 (80–980)
4	*Pratylenchus*	26	240 ± 40 (400–980)
5	*Amplimerlinus*	32	340 ± 60 (200–880)
6	*Helicotylenchus*	40	420 ± 50 (280–780)
7	*Tylenchus*	34	320 ± 40 (200–480)
8	*Pratylenchoides*	30	640 ± 160 (200–980)
9	*Scutylenchus*	28	550 ± 60 (100–860)
10	*Filenchus*	24	280 ± 20 (100–200)
11	*Boleodorus*	20	120 ± 160 (140–360)
12	*Basiria*	16	240 ± 30 (160–900)

* (Nematode infested samples/total samples) × 100. ** Mean nematode abundance/100 g soil ± SD (min and max).

**Table 4 microorganisms-10-01534-t004:** Infestation rate of the sampled fields by *Pratylenchus* spp. in Hendek, Pamukova, Geyve, and Akyazı districts in wheat fields.

No.	District	Number of Fields Surveyed	Infestation (%)	Average Number of Nematodes/100 g *
1	Hendek	5	3	40
2	Pamukova	4	4	20
3	Geyve	7	2	15
4	Akyazı	5	5	25
Total		22	35 *	25

* Infestation based on the total number of samples (40 samples).

**Table 5 microorganisms-10-01534-t005:** Sequenced *Heterodera filipjevi* samples were collected from wheat fields in Sakarya Province.

No.	District	Location	Latitude	Longitude
1	Kaynarca	Kızılcaali	40°98′66″	30°40′71″
2		Kızılcaali	40°98′27″	30°40′58″
3		Merkez	41°03′33″	30°30′77″
4		Küçükkaynarca	41°03′38″	30°32′48″
5	Geyve	Merkez-I	40°50′41″	30°30′55″
6		Merkez-II	40°51′74″	30°31′06″
7		Çamlık	40°45′24″	30°37′93″
8		Ilıca	40°45′25″	30°38′10″
9	Taraklı	Hacıyakup-I	40°44′69″	30°37′94″
10		Hacıyakup-II	40°43′95″	30°43′85″
11		Aksu	40°44′64″	30°45′37″
12	Söğütlü	Merkez	40°89′69″	30°46′43″

**Table 6 microorganisms-10-01534-t006:** Morphological and morphometrical characteristics of second juveniles and cysts of *Heterodera filipjevi* populations (*n* = 10); measurement unit in µm.

Morphological Characters	Geyve	Taraklı	Kaynarca	Söğütlü
Body length	521.3 ± 4.84 ^b,^* (482.8–542.5)	548.3 ± 3.74 ^a^ (471.8–588.6)	518.9 ± 6.62 ^b^ (467.8–548.5)	556.3 ± 7.52 ^a^ (481.5–589.8)
Stylet length	23.2 ± 0.45 ^b^ (22.2–25.6)	24.9 ± 0.32 ^a^ (22.5–27.6)	23.5 ± 0.49 ^b^ (21.6–24.8)	24.2 ± 0.38 ^a,b^ (21.6–25.9)
Tail length	50.45 ± 1.86 ^c^ (43.4–62.6)	52.66 ± 3.35 ^b^ (41.6–64.6)	54.68 ± 3.94 ^a^ (45.6–66.8)	51.24 ± 4.52 ^b^ (42.5–67.6)
Hyaline tail tip length	25.6 ± 1.32 ^b^ (21.4–34.5)	26.6 ± 2.57 ^a^ (22.4–32.3)	25.8 ± 1.45 ^b^ (21.9–37.6)	25.4 ± 1.88 ^b^ (20.4–36.4)
Fenestral length	51.32 ± 2.32 ^b^ (42.1–63.4)	58.34 ± 3.44 ^a^ (43.8–68.4)	54.54 ± 2.32 ^a,b^ (45.2–63.2)	48.34 ± 1.64 ^b^ (40.2–59.8)
Semifenestral width	20.4 ± 2.43 ^c^ (15.2–26.6)	24.4 ± 2 ^a^ (16.7–24.6)	23.2 ± 1.04 ^a,b^ (16.8–27.5)	22.4 ± 2.09 ^b^ (15.2–28.6)
Vulval bridge width	10.56 ± 1.32 ^b^ (6.8–17.2)	14.44 ± 2.43 ^a^ (9.2–18.6)	12.09 ± 2.33 ^a,b^ (7.88–16.54)	15.65 ± 2.33 ^a^ (9.2–17.5)
Vulval slit length	19.5 ± 1.02 ^b^ (12.6–25.8)	22.6 ± 1.58 ^a^ (10.3–26.2)	20.2 ± 2.34 ^a,b^ (13.4–25.6)	18.2 ± 3.32 ^b^ (13.3–25.5)

* Means in the same raw followed by the same letter(s) were not significantly different according to Tukey HSD test (*p* ≤ 0.05).

**Table 7 microorganisms-10-01534-t007:** Diversity of migratory nematodes in wheat field using Shannon diversity index.

Nematode Genus	Average Number of Nematodes/100 g Soil (Frequency)	*pi*	*ln*(*pi*)	*pi* × *ln*(*pi*)
*Merlinus*	960	0.213808	−1.54267	−0.32984
*Trophurus*	240	0.053452	−2.92897	−0.15656
*Paratrophurus*	140	0.031180	−3.46797	−0.10813
*Pratylenchus*	240	0.053452	−2.92897	−0.15656
*Amplimerlinus*	340	0.075724	−2.58066	−0.19542
*Helicotylenchus*	420	0.093541	−2.36935	−0.22163
*Tylenchus*	320	0.071269	−2.64129	−0.18824
*Pratylenchoides*	640	0.142539	−1.94814	−0.27769
*Scutylenchus*	550	0.122494	−2.09969	−0.25720
*Filenchus*	280	0.062361	−2.77482	−0.17304
*Boleodorus*	120	0.026726	−3.62212	−0.09680
*Basiria*	240	0.053452	−2.92897	−0.15656
Total	4490	1	−31.8336	−2.31767
			H = 2.31767	
			E_H_ = 0.932699	

“H” represents the Shannon diversity index; “E_H_” indicates the evenness; “*pi*” indicates the proportion of the entire community made up of species *i*; “*ln*” represents the natural log

## Data Availability

Data generated in this study are available upon reasonable request to the corresponding author.

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
