# Peer review of "Diversity and Identification of Plant-Parasitic Nematodes in Wheat-Growing Ecosystems"

_microorganisms, 2022, doi:10.3390/microorganisms10081534_

Round 1

Reviewer 1 Report

The manuscript entitled “Biodiversity, identification, and molecular phylogeny of plant-parasitic nematodes in wheat-growing ecosystems.” It is an excellent contribution to the scientific community; it shows the importance of studying organisms that can cause losses in food production.

The only doubt is about the reference of the set of primers used to perform the PCR methodology; the manuscript cited is not available. I have reviewed other manuscripts that indeed indicate that they are indeed universal primers that allow the amplification of rDNA-ITS; perhaps the authors could add a recent citation to fill in this slight gap.

Author Response

Dear Editor in Chief,

On behalf of the authors and myself I would like to thank you and the reviewers for your valuable comments on the manuscript. We did all corrections and concerns kindly raised by the reviewers and included in the manuscript. All the corrections made were colored in yellow in the manuscript. English was also improved as suggested by the reviewers. I hope with this corrections the manuscript is improved and now considered to be accepted in the Microorganisim Journal.

Reviewer #1: The only doubt is about the reference of the set of primers used to perform the PCR methodology; the manuscript cited is not available. I have reviewed other manuscripts that indicate that they are universal primers that allow the amplification of rDNA-ITS; perhaps the authors could add a recent citation to fill in this slight gap.

  • Dear Reviewer #1, thank you for your valuable comment. A recent citation was added addressing your comments as following AB28 (5ʹ-CGTAACAAGGTAGCTGTAG-3ʹ) and reverse primer TW81 (5ʹ-TCCTCCGCTAAATGATATG3ʹ) (Joyce et al., 1994).

Reviewer 2 Report

The paper is dealing with plant-nematodes associated with wheat fields in Sakarya region in Turkey. The study is important since plant parasitic nematodes are responsible for substantial loss in cereals  world-wide and knowledge on their presence and population densities could help for their effective management. However, the manuscript needs considerable improvement.  I have some remarks and suggestions below and in the text.

Title

The title is somewhat misleading since phylogeny is provided only for one species - Heterodera filipjevi.

Introduction

The introduction should be extended and data for other plant parasitic nematodes occurring in wheat fields from other regions including Turkey to be considered. The objectives of the study should be more clearly outlined.

Material and methods

Literature sources used for plant nematodes identification should be mentioned.

I suggest the authors to provide for each field from where samples have been collected some more data – the type of management, previous crop if rotations are applied, use of pesticides etc. It will facilitate interpretation of results.  

And it is not clear to me how many samples have been collected – it is said that each sampling site was visited 3 times -e.g. 10 samples for migratory and 5 samples for sedentary nematodes are 15 samples for each locality, not 40.

Results

This section should be better structured, assessment of diversity is better to follow the results of occurrence of plant-parasitic nematodes and a table with H values for each site to be provided. The results should be more clearly presented.

I wonder how no members of rhabditids, cephalobids or panagrolaimids have been detected.

Discussion

I think that part of this section should go to the introduction part. Also, the discussion should be more focused.

The manuscript is not carefully prepared, it needs a considerable revision.

Author Response

Dear Editor in Chief,

On behalf of the authors and myself I would like to thank you and the reviewers for your valuable comments on the manuscript. We did all corrections and concerns kindly raised by the reviewers and included in the manuscript. All the corrections made were colored in yellow in the manuscript. English was also improved as suggested by the reviewers. I hope with this corrections the manuscript is improved and now considered to be accepted in the Microorganisim Journal.

Reviewer #2: The title is somewhat misleading since phylogeny is provided only for one species - Heterodera filipjevi.

  • Dear Reviewer #2, The title was changed to” Biodiversity and identification of plant-parasitic nematodes in wheat-growing ecosystems”.

Reviewer #2: The introduction should be extended and data for other plant parasitic nematodes occurring in wheat fields from other regions including Turkey to be considered. The objectives of the study should be more clearly outlined.

  • Dear Reviewer #2, A paragraph was added to address your comments.

Plant-parasitic nematodes alone are responsible for nearly 10% of yield losses worldwide, resulting in economic losses of over $125 billion per year [4]. However, yield losses associated with PPNs generally go unnoticed by a great proportion of farmers due to the soil-dwelling nature of most PPNs and the similarity of symptoms with other biotic and abiotic disease-inducing factors. Even though it is established that the problem is nematode-originated, it is still quite difficult to differentiate PPN species due to the co-occurrence of different species of PPNs in the field and complexity of diagnostic characteristics. Among PPNs that decrease the yield and profitability, Cyst (Heterodera spp.), Root-lesion (Pratylenchus spp.), Root-knot (Meloidogyne spp.), Seed-gall (Anguina spp. ) and Stubby-root nematodes (Paratrichodorus spp.) are the most economically important genera [5].

[5]. Smiley, R. W., & Nicol, J. M. (2009). Nematodes which challenge global wheat production. Wheat science and trade, 171-187.

Reviewer #2: Literature sources used for plant nematodes identification should be mentioned.

  • Dear Reviewer #2, Literature sources used for the identification of plant nematodes were mentioned (number 35,36, 37).

Reviewer #2: I suggest the authors to provide for each field from where samples have been collected some more data – the type of management, previous crop if rotations are applied, use of pesticides etc. It will facilitate interpretation of results.

  • Dear Reviewer #2, Table 1 contains information on samples that have been collected from which district and location. All samples were collected in monoculture cereal production areas, and this statement was added to the manuscript. Also, there was no use of pesticides for soil-borne diseases.

Reviewer #2: And it is not clear to me how many samples have been collected – it is said that each sampling site was visited 3 times -e.g. 10 samples for migratory and 5 samples for sedentary nematodes are 15 samples for each locality, not 40.

  • Dear Reviewer #2, A total of 40 soil samples were taken from each sampling location for each sampling time (wheat's heading flowering and harvest time). We added “for each sampling time” at the end of the sentence to avoid misunderstanding.

A total of 40 soil samples were randomly collected from each sampling location for each sampling time (Table 1).

Reviewer #2: Results This section should be better structured; an assessment of diversity is better to follow the results of occurrence of plant-parasitic nematodes and a table with H values for each site to be provided. The results should be more clearly presented.

  • Dear Reviewer #2, Each site is in Sakarya province. Therefore, to show a clear picture, H values of migratory nematode genera recovered from the sampled fields in Sakarya province were given.

Reviewer #2: I wonder how no members of rhabditids, cephalobids or panagrolaimids have been detected.

  • Dear Reviewer #2, It was thought that the absence of some nematode species/genus might be related to the variety and lines of the crops, the fact that the crops have been planted in the same field for years, and/or whether they are poor hosts of nematodes.

Discussion

Reviewer #2: I think that part of this section should go to the introduction part. Also, the discussion should be more focused.